# Yeast Genomics and Its Applications in Biotechnological Processes: What Is Our Present and Near Future?

**DOI:** 10.3390/jof8070752

**Published:** 2022-07-20

**Authors:** Vivian Tullio

**Affiliations:** Department Public Health and Pediatrics, Microbiology Division, University of Turin, Via Santena 9, 10126 Torino, Italy; vivian.tullio@unito.it

**Keywords:** *Saccharomyces cerevisiae*, genomics, genetic manipulation, production of protein heterologues, vaccines, biofuels, yeast diary products

## Abstract

Since molecular biology and advanced genetic techniques have become important tools in a variety of fields of interest, including taxonomy, identification, classification, possible production of substances and proteins, applications in pharmacology, medicine, and the food industry, there has been significant progress in studying the yeast genome and its potential applications. Because of this potential, as well as their manageability, safety, ease of cultivation, and reproduction, yeasts are now being extensively researched in order to evaluate a growing number of natural and sustainable applications to provide many benefits to humans. This review will describe what yeasts are, how they are classified, and attempt to provide a rapid overview of the many current and future applications of yeasts. The review will then discuss how yeasts—including those molecularly modified—are used to produce biofuels, proteins such as insulin, vaccines, probiotics, beverage preparations, and food additives and how yeasts could be used in environmental bioremediation and biocontrol for plant infections. This review does not delve into the issues raised during studies and research, but rather presents the positive outcomes that have enabled several industrial, clinical, and agricultural applications in the past and future, including the most recent on cow-free milk.

## 1. Introduction

Yeasts are well-known fungi that have been used by humans since ancient times to meet the food needs of populations. From bread to beer, the production of sauerkraut, and genetic engineering, etc., they represent a very heterogeneous group of organisms within the same species, including from a genetic standpoint. The term “yeast” has no taxonomic meaning; it refers to a group of morphologically unicellular microorganisms that are oval shaped and approximately 10 microns in size, mostly colorless or white in color, and are so-called because they cause the “bread to rise or peach juice to rise or in making alcoholic drinks such as beer”. However, their shape is not stable, as they can become filamentary, forming hyphae similar to multicellular filamentous mycetes, which can then become yeast in certain circumstances. What is causing yeast to transform in this manner? Certainly, this morphological variability is due to the extensive genetic makeup of eukaryotes. To understand what yeasts are, their genetic makeup, and the future prospects for their use, it is necessary to first clarify the evolutionary history of fungi and their classification [1].

### What Are Fungi?

The first question that arises when we talk about fungi is that which concerns their evolutionary location: up to approximately 40–50 years ago, they were considered to be vegetables and inserted by botanists among the Cryptogams (without flowers) or between the Thallophytes, as in the fungi. According to the classification of Linnaeus, there is no difference between roots, stems, and leaves, as in the Cormophytes. Microbiologists, on the other hand, considered them superior Protists, but since 1969, they have been included in a separate kingdom: Fungi (Mycota). The reason for this decision derives from the fact that fungi have characteristics in common with both plants and animals and, of course, their own characteristics. However, recent molecular biology studies have shown that fungi are more closely related to animals than to plants [1].

Fungi have very ancient origins, with evidence showing their appearance on Earth about 1 billion years ago [2]. Fossil fungi are difficult to find and study due to their perishable structure, but organisms recognizable as fungi (remarkably similar to modern species) appeared as early as the Ordovician period about 450 million years ago in the form of lichens, plant parasites, and fungal and mycorrhiza-like associations. A very peculiar fossil is the Late Silurian Prototaxites, which was originally considered a tree but the most recent analysis revealed that it was a fungus capable of growing up to 8 m in height [1]!

Fungi and plants have shared the majority of their evolutionary history. Plants would not have been able to colonize the earth if it had not been for fungi. The earliest rootless land plants are thought to have evolved from freshwater algae that formed close associations with filamentous fungi to obtain water and minerals from arid substrates such as nitrogen and phosphorus. Plant–fungi interactions would have allowed terrestrial plants to dominate continents, transforming the lithosphere (the earth’s outer shell), biosphere (the earth’s living systems), and atmosphere into what they are today [1,3].

The Fungi kingdom contains a wide range of species, with approximately 144,000 known species, although its number is steadily increasing as approximately 2000 new species are discovered and identified each year [4].

The vast majority (over 93 percent) of fungal species are currently unknown, and the total number of species on Earth is estimated to be between 2.2 and 3.8 million, a number that exceeds that of plants by more than six times [4,5].

In 2007, 536 families of mycetes had been recognized and identified. About ten years later, in 2017, there was talk of eight fungal phyla and, just one year later, in 2018, nine sub-kingdoms and no less than 18 phyla with more than 700 families. This reflects the large number of new taxa recognized each year, mainly as a result of the recent rapid increase in the availability and efficiency of DNA sequencing-based molecular biology methods for the detection and identification of fungi. Recently, some intracellular parasites of microscopic algae, considered to be more closely related to animals and historically named according to animal nomenclature, have been reclassified and included in a fungal subregion (Aphelidiomyceta). It is still too early to know whether or not to accept these new classifications, and further studies could lead to further upheavals. However, they highlight how rapidly our understanding of what it means to “be a fungus” is changing and how new discoveries are reevaluating their classification [1,3].

While the traditional classification of fungi was based solely on morphological (moulds) and physiological (yeasts) characteristics that did not necessarily reflect evolutionary history, the modern classification of fungi is mainly based on groups defined by the common ancestry of their DNA sequences, with other characteristics providing evidence in support. Due to difficulties in distinguishing them using morphological characters, a number of species that were once “grouped” are now considered to be distinct species (so-called cryptic species because they appear identical) [6]. Thanks to molecular biology and DNA analysis, e.g., genes for ribosomal RNA (ITS) or sequences of the 18S-rRNA gene, which are overcoming traditional classification schemes, it was found that not all fungi with similar structures evolved from the same ancestral lines (convergent evolution). However, molecular analysis is not free from problems related mainly to the reproducibility of the samples, especially environmental ones, to the ability to discriminate between different families and to the lack of a precise nomenclature to follow to name known species only on the basis of the analysis of DNA [6].

The deeper we go into the knowledge of fungi, the more we realize how this intriguing kingdom of organisms is the basis of all life on earth. Not only they are essential for the decomposition of matter and for the recycling of nutrients, but they also provide many benefits for humans. Just think of edible mushrooms and truffles, the production of bread and beer by yeasts, the production of cheeses by filamentous fungi, and the production of antibiotics, enzymes, and organic acids, etc. [1].

Because of this potential, as well as their manageability, safety, and ease of cultivation and reproduction, yeasts are now being studied extensively in order to evaluate a growing number of natural and sustainable applications.

This review will describe what yeasts are, how they are classified, and attempt to provide a rapid overview of the many current and future applications of yeasts. The review will then discuss how yeasts, including those molecularly modified, are used to produce biofuels, proteins such as insulin, vaccines, probiotics, beverage preparations, and food additives, and how they could be used in environmental bioremediation and biocontrol for plant infections. This review presents the positive outcomes that have enabled several industrial, clinical, and agricultural yeast applications in the past and future, including the most recent on cow-free milk.

## 2. Yeasts and Their Genomic Sequences

On 24 April 1996, the baker’s yeast *Saccharomyces cerevisiae* laboratory strain S288c was announced to be the first eukaryote to have its entire genome sequence, consisting of 12 million base pairs, fully sequenced as part of the Genome Project [7,8]. At the time, it was the most complex organism to have its full genome sequenced, and the work took seven years and the involvement of more than 100 laboratories [8,9]. Because of knowledge of its structure, genome, metabolism, and other factors, *S. cerevisiae* has received the most attention among microbial species and is one of the most widely used microorganisms in a wide range of applications, from ecological scientific research to commercial and industrial applications [10,11,12]. Through fermentation, this yeast converts carbohydrates to carbon dioxide and alcohol. For thousands of years, the products of this reaction have been used in baking and the production of alcoholic beverages [13]. *Saccharomyces cerevisiae*, a Hemiascomycetous yeast, is an important model organism in modern cell biology research and one of the most extensively studied eukaryotic microorganisms [14]. Researchers cultured this yeast to learn more about the biology of the eukaryotic cell and, ultimately, human biology [15]. Yeasts have recently been used to produce ethanol for the biofuel industry and to generate electricity in microbial fuel cells [13,16,17].

Genome sequences from an increasing number of yeast species have become available in recent years, resulting in significant advances in the understanding of the evolutionary mechanisms in eukaryotes. Among eukaryotic organisms, yeasts provide unique advantages for evolutionary genomic studies. These unicellular fungi are easily amenable to microbial genetic techniques due to their ability to perform unlimited clonal propagation by budding or fission and the limited size and compactness of their genomes facilitate the characterization of naturally or artificially evolved populations using sequencing [18,19]. Contrary to popular belief, yeasts are not primitive unicellular eukaryotes, but rather have repeatedly emerged from distinct phylogenetic lineages of “modern” fungi. *Saccharomyces cerevisiae* is an unrivaled resource for studying basic molecular mechanisms in eukaryotic cells, as more than 80% of its 5780 protein-coding genes have been functionally characterized [18].

*Schizosaccharomyces pombe* (also known as “fission yeast”) is a yeast species used in traditional brewing and as a model organism in molecular and cell biology. Its genome, which is approximately 14.1 million base pairs in length, was the second to be completely sequenced in 2002 [20], and it is considered to contain 4970 protein-coding genes and at least 450 non-coding RNAs [21]. Because this yeast is only distantly related to *S. cerevisiae* and its genetic architecture is very different (i.e., *S. cerevisiae* has 16 chromosomes, *S. pombe* has 3 and *S. cerevisiae* is often diploid, while *S. pombe* is usually haploid, etc.), comparing the two species did not yield many interpretable observations in terms of evolutionary genomics [22]. The extensive sequence divergence observed between different yeast lineages is a major and unexpected lesson from yeast genomics. This divergence extends all the way down to the species level and reflects intense genomic changes, which contrast with yeasts’ long-term conservation of biological properties [23].

Genomes from different yeast clades, or even species in the same genus, diverge abruptly from one another, rather than giving a continuous spectrum of incremental evolutionary adaptations as indicated by conventional Darwinian theory. This is consistent with yeast genomes being the leftovers of multiple bottleneck occurrences in essentially clonal populations. The random drift that results from this mode of dissemination is significant because it allows nonoptimized variants to survive and eventually colonize novel niches where they may be better adapted [18,24].

Under suitable conditions, all yeasts can generate indefinitely through mitotic divisions, resulting in huge clonal populations of either the haploid or diploid phases, or even polyploids. One of the most fundamental requirements for distinguishing yeasts from other fungi is their capacity for infinite clonal development in unicellular form, paired with the fact that their sexual states do not create fruiting bodies [10,18]. Multiple individual genome sequences are needed to understand the forces shaping sequence variation and the relationship between genotype and phenotype. Furthermore, determining the link between genotype and phenotype within a community necessarily requires a genome-wide examination of polymorphism patterns in a large sample of individuals [10,14,18].

Our limited grasp of the variation within yeast species contrasts with the plethora of evidence on interspecies sequencing differences. Nonetheless, the last decade has seen tremendous advances in describing polymorphic variation within yeast species. Yeast population genomics has so far mainly focused on two species: *S. cerevisiae* and *S. paradoxus* [25]. Some studies have looked into the genetic diversity of huge groups of yeast strains belonging to the same species [10,26].

This newfound understanding of not just what each gene and gene product accomplishes for the organism and how all of these genes, gene products, activities, and their regulation interact to form the body’s characteristics has sparked a new wave of genomic science. As a result, this novel “system-level” biological scientific research [27,28] tries to explain how gene interactions can control and maintain homeostasis throughout the life cycle of an organism exposed to a variety of stimuli and environmental factors. As a result, *S. cerevisiae* has become a forerunner in biology’s new era [10,14].

## 3. Genetically Engineered Biofactories

*Saccharomyces cerevisiae* and *S. pombe* have been widely used in genetics and cell biology because they are simple eukaryotic cells that serve as models for all eukaryotes, including humans. Besides that, they allow researchers to investigate fundamental cellular processes such as the cell cycle, DNA replication, recombination, cell division, and metabolism. Furthermore, because yeasts are easily manipulated and cultured in the laboratory, powerful standard techniques such as yeast two-hybrid [29], synthetic genetic array analysis [30], and tetrad analysis have been developed. Many proteins important to human biology were discovered through study of their yeast homologues, including cell cycle proteins, signaling proteins, and protein-processing enzymes [31]. As of 2014, the genomes of over 50 yeast species had been sequenced and published [32].

Because of its use in fossil fuels and environmental degradation, microbiology is seen as a possible alternative for the synthesis of important compounds due to its mild reaction conditions and environmentally acceptable properties. *S. cerevisiae* is the yeast that has gotten the most attention among microbial species and it is one of the most widely used microorganisms in a wide range of applications from scientific research to commercial and industrial applications, thanks to knowledge of its structure, genome, metabolism, and other factors [33,34].

Various yeast species have been genetically engineered to produce various drugs more efficiently, a technique known as metabolic engineering [35,36,37]. *Saccharomyces cerevisiae* is simple to genetically engineer; its physiology, metabolism, and genetics are well understood, and it can be used in harsh industrial environments. Engineered yeast can produce a wide range of chemicals from various classes, including phenolics, isoprenoids, alkaloids, and polyketides [37]. *Saccharomyces cerevisiae* produces approximately 20% of biopharmaceuticals, including insulin, hepatitis vaccines, and human serum albumin [38].

## 4. Biofuel Production

Because of the high consumption of fossil fuels and environmental pollution, microbial biotechnology is seen as a promising alternative for valuable chemical production due to its mild reaction conditions and environmentally friendly properties [39,40].

The ability of yeast to convert sugar into ethanol has been used by the biotechnology industry to produce ethanol fuel. The first step is to mill a feedstock like sugar cane, field corn, or other cereal grains, followed by the addition of dilute sulfuric acid or fungal alpha amylase enzymes to break down the starches into complex sugars. The complex sugars are then broken down into simple sugars by glucoamylase. Because *S. cerevisiae* spontaneously produces ethanol by converting one mole of glucose (180 g) into two moles of ethanol (92 g) plus two moles of carbon-dioxide (88 g) and energy (26.4 Kcal), yeast is added to convert the sugars to ethanol, which is then distilled to produce ethanol with up to 96 percent purity [16,40] *Saccharomyces* spp. have been genetically engineered to ferment xylose, a major fermentable sugar found in cellulosic biomasses like agricultural residues, paper waste, and wood chips [41,42]. As a result of this breakthrough, ethanol can now be produced more efficiently from lower-cost feedstocks, making cellulosic ethanol fuel a more competitively priced alternative to gasoline fuel [40,43].

Many *S. cerevisiae* strains have been genetically manipulated or metabolically engineered to produce higher alcohols such as 1-butanol, isobutanol, farnesene, and bisabolene (sesquiterpenes), as well as biodiesel (naturally occurring fatty acid ethyl esters) [44,45]. n-butanol or 1-butanol, a four-carbon alcoholic molecule, as well as isobutanol and isopropanol, have demonstrated a number of physical characteristics that distinguish them as fuels from ethanol. One of 1-butanol’s physical properties is that it is less hygroscopic, less corrosive, and has a higher energy density and octane value. As a result, n-butanol can be mixed with gasoline in almost any ratio [40].

Metabolic engineering techniques were used to increase butanol production in *S. cerevisiae* and the synthetic acetone–butanol–ethanol (ABE) route increased butanol production while simultaneously establishing butanol resistance [46]. Isobutanol is an important next-generation biofuel that has gotten a lot of attention due to its biological production from yeast and use as a component in basic chemical manufacturing. It has been demonstrated that metabolically engineered strains of *S. cerevisiae* that produce isobutanol have advantageous characteristics that make isobutanol production cost effective [40].

The compound 1-propanol is a three-carbon main alcohol. It has a high octane rating and is suitable for use in engines. However, the cost of generating propanol has been assessed as too expensive for it to be a common fuel. As a result, an appropriate microbial strain was chosen for 1-propanol production. The yeast strain had a considerable influence on n-propanol, but not so much on the other alcohols [40].

Isopropyl alcohol, also known as 2-propanol, is a colorless, moderately volatile alcohol with a strong characteristic odor that is not unpleasant when pure. It is frequently referred to as “isopropanol”, despite the fact that the name is incorrect according to IUPAC. In fact, if it had such a name, isopropyl alcohol would have to derive “from isopropane”, a nonexistent hydrocarbon, because propane clearly has no isomers. However, the name has become popular as this alcohol is a constitutional isomer of propanol (n-propane). A *Candida utilis* yeast strain that has been genetically modified has been developed for the production of isopropanol, which has potential industrial and home applications as an anti-bacterial, disinfectant, and detergent component [47]. Using low-cost fermentation technologies, these alcohols would be produced as a biofuel component or as a precursor for the chemical synthesis of propylene [40,48].

Lactic acid (LA), succinic acid (SA), and ethylene are three additional commercial products of yeast biosynthesis that are produced in large quantities, ranging from 10 to 180 kilotons per year. Furthermore, because of its well-understood and well-characterized lipid metabolism, *S. cerevisiae* may be a promising host for polyunsaturated fatty acid synthesis due to the availability of genetic tools and an efficient fermentation technique. *Saccharomyces cerevisiae* has recently been engineered to produce short-chain or branched-chain fatty acids [49,50]. Engineered yeast, on the other hand, contributes to the production of fatty acid-derived oleochemicals and biofuels, a promising development for future biofuel production [40].

*Meyerozyma guilliermondii* [51] is a prospective biotechnology species that can only thrive on n-alkanes. It is one of the yeasts that can biotransform xylitol, a natural and healthful sweetener [52,53]. This yeast is also a model organism for riboflavin overproduction and the production of industrial enzymes such as inulinase and lipase. This yeast-mediated biocontrol has definitely emerged as a promising approach for preventing and decreasing rot loss in harvested products, an alternative to the application of fungicide. When compared to other well-studied yeasts like *Pichia pastoris* and *S. cerevisiae*, *M. guilliermondii* has various advantages in biotechnology processes [53].

## 5. Food Industry

In 1920, the Fleischmann Yeast Company launched a “Yeast for Health” campaign to promote yeast cakes. Initially, they emphasized yeast as a source of vitamins that are beneficial to the skin and digestion. Their later advertising claimed a much broader range of health benefits, which the Federal Trade Commission called misleading. Yeast cakes were popular until the late 1930s [54]. People are becoming increasingly conscious of the nutritional benefits of foods as well as their composition and origin, and they prefer to consume certified and high-quality foods with distinct tastes and flavors. As a direct consequence, yeasts have another potential application in the food industry, and several yeasts including *Kluveryomyces*, *C. kefir*, *C. palmioleophila* (*Torulopsis candida)*, *Pichia* spp., *S. boulardii* are employed in the dairy industry to make cheese, kefir, and koumiss [55]. Cremont is a mixed-milk cheese made by combining cow and goat milk with “Vermont cream” [40]. It is blended together and pasteurized, then a special cocktail of yeasts (*K. marxianus*, *P. deserticola*, *P. fermentans*, *P. manshurica*, *P. membranaefaciens*, and *S. cerevisiae*) and mold (*Geotrichum candidum*) is added to create its unique flavor and naturally coagulate the milk overnight. The next day, fresh curd is shaped by hand into cylinders [40]. Aesthetics and aromas come from the *G. candidum* that grows on the surface of cheeses and looks like tiny single cells that look like a yeast [40].

### 5.1. Nutritional Supplements

Yeast is used in foods for its umami flavor similar to monosodium glutamate (MSG), and, like MSG, often contains free glutamic acid [40]. Yeast extract is a food additive or flavor made from the intracellular contents of yeast [56,57]. On a commercial scale, the general method for producing yeast extract for food products such as Vegemite and Marmite is heat autolysis, which involves adding salt to a suspension of yeast to make the solution hypertonic, causing the cells to shrivel up [40,56,57]. This results in autolysis, a self-destructive process in which the yeast’s digestive enzymes degrade their own proteins into simpler compounds. The dying yeast cells are then heated to complete decomposition and the husks (yeast with thick cell walls that produce poor texture) are removed. On a commercial scale, yeast autolysates are used to make salty paste primarily used as a spread on sandwiches and toast. These food are manufactured in many countries under different names but with quite similar characteristics: Vegemite and Promite (Australia), Marmite (UK and New Zealand), Vitam-R (Germany), and Cenovis (France and Switzerland) [58,59].

Nutritional yeast is whole dried, deactivated yeast cells, most commonly *S. cerevisiae*. Its nutty and umami flavor makes it a vegan substitute for cheese powder and is typically found in the form of yellow flakes or powder [60]. Another common application is as a topping for popcorn. It is also good in mashed and fried potatoes as well as scrambled eggs. It is available as flakes or as a yellow powder with a texture similar to cornmeal. It is sometimes referred to as “savoury yeast flakes” in Australia [60].

Both of the yeast foods mentioned above are added with B-complex vitamins [60], making them an appealing vegan nutritional supplement. Nutritional yeast, in particular, is naturally low in fat and sodium, as well as a source of protein, vitamins, and other minerals and cofactors needed for growth.

*Lactobacillus acidophilus* fermented milk is a beverage created by fermenting milk in conditions that favor the growth and development of a large number or kind of organism by the bacterial strain *L. acidophilus*. To improve the flavor and the antioxidant qualities of this beverage and the survivability of bacterial strains, yeast strains such as *S. boulardii* and *S. cerevisiae* have been introduced to this fermented milk. Lactic acid (1.5 to 2.0%) is present in this beverage, which aids in the therapy of a patient’s gastrointestinal condition [40,61].

### 5.2. Nonalcoholic Beverages

A variety of sweet carbonated beverages can be produced using the same methods as beer, with the exception that the fermentation is stopped sooner, resulting in carbon dioxide but only trace amounts of alcohol, leaving a significant amount of residual sugar in the drink. Native Americans invented root beer, which Charles Elmer Hires popularized in the United States during Prohibition [62].

Kvass is a popular fermented rye drink in Eastern Europe. It has a distinct flavor but is low in alcohol and contains B-complex vitamins such as the above-mentioned yeast-fermented products [63].

Kombucha is a fermented tea that has been sweetened. It is produced using yeast and acetic acid bacteria. Some of the yeast species found in tea include *Brettanomyces bruxellensis*, *C. stellata, S. pombe*, *Torulaspora delbrueckii*, and *Zygosaccharomyces bailii*, known as chajnyj grib (Russian: aн pи) in Eastern Europe and some former Soviet republics. Numerous implausible health benefits have been claimed to correlate with this drink. However, there is little evidence to support any of these claims. In addition, since it is usually prepared at home, the beverage has caused rare serious adverse effects, possibly arising from contamination during home preparation [64].

Milk is fermented with yeast and bacteria to make kefir [65] and kumis [10,55]. Kefir is a fermented milk drink made from kefir grains. In these grains, a complex and highly variable symbiotic community can be found, which can include acetic acid bacteria, yeasts (such as *C. kefyr* and *S. cerevisiae*), and a number of *Lactobacillus* species, such as *L. parakefiri*, *L. kefiranofaciens* (and subsp. *kefirgranum*). The drink originated in the North Caucasus, from where it spread to Russia and from there to Europe and the United States, where it is made by inoculating cow, goat, or sheep milk with the kefir grains [65].

Kumis is a fermented dairy product traditionally made from mare or donkey milk. The Turkic and Mongol peoples of the Central Asian steppes still drink it [10,55]. Kumis is a dairy product similar to kefir, but it is made from a liquid starter culture rather than solid kefir “grains”. As mare’s milk contains more sugars than cow or goat milk, kumis has a higher, though still mild, alcohol content when fermented than kefir [66].

Mauby is made by fermenting sugar with wild yeasts found on the bark of the Caribbean trees *Colubrina elliptica* and *C. arborescens*, which are native to the northern Caribbean and south Florida. Other ingredients are usually included in recipes, with spices such as aniseed being very common. Mauby used to be a fermented beverage made in small batches, but it is now mostly a commercial non-fermented soft drink [67].

## 6. Biopharmaceutical Production: Proteins and Vaccines

Yeasts play an important role in the clinical field by producing heterologous proteins that are used for therapeutic purposes [68]. These biopharmaceutical therapeutic proteins have been known since the 1980s and are produced by *S. cerevisiae* using modern biology techniques (e.g., genetic manipulation and monoclonal antibodies produced using hibridoma technology). Insulin and its analogues produced using *S. cerevisiae* are one example of a significant advance in the treatment of diabetes. Moreover, a large number of companies use *S. cerevisiae* to produce a wide range of recombinant proteins, including those used to treat diabetic ulcers, uricaemia, thrombocytopenia, and prevent thrombosis [40,69].

The production of heterologous proteins by yeast is a crucial step in the industrial manufacture of enzymes and biopharmaceuticals. Genetic and process engineering solutions have been utilized to address the challenges. *P. pastoris* is a methylotrophic yeast and a “usually considered as harmless” microbe. As a result of the increased demand in these industries, *P. pastoris* was utilized to manufacture over 500 medicinal proteins and over 1000 recombinant proteins, as well as enzymes like xylanase and phytase which are relevant to the food and feed industries [70].

Another application of yeasts is in the development of modern and safe vaccines using recombinant DNA technology [71,72]. Traditional techniques may have limitations such as the reversal of the live attenuated vaccine strain to a virulent wild strain, which is rare but still possible, or in the case of killed/inactivated vaccines, a minimum residual toxicity in the final formulation which can be overcome with recombinant DNA.

Furthermore, conventional vaccines necessitate a mass culture of associated or related organisms as well as lengthy incubation periods. The special requirements for storage and transportation raise the cost of vaccine preparations even further. Vaccines produced with recombinant DNA techniques are usually made by cloning and the expression of genes encoding vaccine immunogenic proteins in bacterial (e.g., *Escherichia coli*) or eukaryotic (yeast) cells that produce large amounts of the vaccine’s heterologous protein component when replicated. Alternatively, the primary structure of the antigen protein can be determined, the nucleotide sequence of its mRNA can be traced, and a copy of this gene can be synthesized [73,74]. As a result, these vaccines are made up of protein components that represent specific virulence factors of the pathogenic microorganism, hence the names “subunit” or “acellular” vaccines. Hepatitis B was the first recombinant vaccine approved for human use (HBV). It is a subunit vaccine produced by cloning and expressing the Hepatitis B surface antigen (HBsAg) in *S. cerevisiae* cells using a plasmid [75]. In this way, vaccines can be developed without requiring the use of human plasma samples. The recombinant antigen is purified (via centrifugation, chromatography, and fractional precipitation) and then absorbed with aluminum hydroxide before being included in the vaccination preparation [73,74].

Many other antibacterial and antiviral vaccines, such as those against hepatitis A, diphtheria, tetanus, pertussis, and *Hemophilus influenzae*, are prepared through genetic manipulation of *S. cerevisiae* [40].

Yeast-based protein production is important in the pharmaceutical industry, accounting for approximately 20% of bioproducts approved for human use, including prophylactic vaccines. It has several advantages over mammalian-based systems, including rapid growth rates, ease of genetic engineering, faster cell line development, low and locally available medium requirements, and well-established fermentative technologies—some of which are similar to those used in the brewing industry. As a result, yeast may be more appropriate for vaccine production in low-resource environments. As a vaccine manufacturing platform, yeast can generate virus-like particles (VLPs) by expressing one or more viral capsid proteins which can form and display a diverse set of antigenic sites, including discontinuous epitopes [76]. While retaining high immunogenicity, VLPs are non-infectious and far safer than live or inactivated vaccine approaches. Currently, VLP-based vaccines are being developed and are available. Several yeast species other than *S. cerevisiae* have been used in the development of human vaccines, including non-conventional yeasts such as *Hansenula polymorpha* (*P. angusta*) and *P. pastoris*. However, since no single yeast platform is ideal for producing all vaccines, several yeast strains are tested in parallel for their ability to produce a specific vaccine with the desired yield and quality [74].

Edible vaccines produced in plants are another intriguing approach in the development of next generation vaccines. Plants are excellent systems for the production of heterologous proteins for vaccination use because they have several advantages: they are resistant to infections by animal pathogens; they are composed of eukaryotic cells capable of carrying out post-translational protein modifications, such as glycosylation; they are edible and thus easy to administer; and they can induce mucosal immunity. This method involves the direct expression of antigens in transgenic plants via infection with the engineered bacterium *Agrobacterium tumefaciens* or the biolistic method. Alternatively, the antigen is transiently expressed in the plant via infection with a recombinant virus, such as the Tobacco Mosaic virus. To date, numerous transgenic plants, including tomatoes, potatoes, bananas, lettuce, and spinach, have been developed to express vaccine antigens such as HBV HBsAg, rabies virus glycoproteins, etc. [77,78].

Reverse vaccinology, an innovative technique developed in the late 1990s by Italian scientist Rino Rappuoli based on the sequencing of the pathogenic microorganism’s genome, has contributed significantly to the development of new vaccines. This technology has become the gold standard for all protein vaccines, allowing for the development of the first recombinant vaccine against group B *Neisseria meningitidis* [79].

## 7. Probiotics and Postbiotics

“Probiotics” are defined by the World Health Organization as “live microorganisms that, when administered in adequate amounts, confer a health benefit on the host”. Scientific evidence supports a wide range of probiotic health benefits, including the role of yeasts in the prevention and treatment of intestinal disorders as well as an improvement in mineral bioavailability via phytate hydrolysis, folate production, and mycotoxin detoxification, all of which are mediated by yeasts [80]. Bacteria, specifically lactobacilli and bifidobacteria, are typically the best probiotics, but *S. boulardii* (fundamental yeast probiotic), is also considered probiotic due to its beneficial properties on intestine function and its ability to stimulate the immune system [81]. In fact, some probiotic supplements contain *S. boulardii*, which helps to maintain and restore the natural microbial population in the gastrointestinal tract. *Saccharomyces boulardii* has been shown to reduce the symptoms of acute diarrhea [82,83], the risk of infection by *Clostridium difficile* [84], reduce bowel movements in diarrhea-predominant IBS (irritable bowel syndrome) patients [85,86], as well as the incidence of traveler’s, antibiotic-, and HIV/AIDS-related diarrhea [83]. *Kluyveromyces*, *Debaryomyces*, *Candida*, *Pichia*, *Hanseniaspora*, and *Metschnikowia* were also discovered to have probiotic potential [80]. Finally, yeast probiotics are still being studied for their anti-cancer biotherapeutic properties, with researchers investigating their important role in cancer prevention and treatment [87]. New research suggests that probiotics may help control the progression of various types of gastrointestinal cancer. Some authors investigated the potential of probiotics in controlling cancer progression, specifically, the slowing of tumor formation and subsequent metastases [88,89,90]. Probiotics and postbiotics (microbial waste molecules, i.e., vitamins B and K, short-chain fatty acids, antimicrobial molecules, etc.) are thought to influence a variety of metabolic pathways in cells, including proliferation, apoptosis, infammation, angiogenesis, and metastasis [88,89,90]. Both in vitro and in vivo, probiotics have been shown to improve the activation of gastrointestinal enzymes, the inhibition of carcinogenic agents, and the suppression of pre-cancerous lesions [89,90].

## 8. Bioremediation

Bioremediation is a biotechnology branch that uses living microorganisms such as mycetes and bacteria to remove contaminants, pollutants, and toxins from soil, water, and other environments. Bioremediation can be used to clean up polluted groundwater or environmental issues like oil spills. Some yeasts have the potential to be useful in bioremediation. The yeast *Yarrowia lipolytica* has been shown to degrade palm oil mill effluent, TNT (2, 4, 6-trinitrotoluene; an explosive material), and other hydrocarbons such as alkanes, fatty acids, fats, and oils [91]. It can also withstand high salt and heavy metal concentrations and is being investigated for its potential as a heavy metal biosorbent [92,93]. Furthermore, *S. cerevisiae* has the potential to bioremediate toxic pollutants found in industrial effluent, such as arsenic, and it has been extensively researched as a biosorbent for a variety of heavy metals, including Pb, Cr, Zn, Cu, and Cd [94]. This microorganism-based method has numerous benefits including low operating costs, a smaller volume of sludge produced, and a high efficiency in detoxifying very dilute effluents [95,96]. Although the term “biosorption” is commonly applied to non-living biomaterials that bind and concentrate contaminants, this process occurs in both living and dead [97]. Similarly, when expressed in *S. cerevisiae*, the gene PtMT2b from the tree species *Populus trichocarpa* cv. “Trichobel” was able to reduce Cd toxicity [94]. Both free and complex silver ions are bioaccumulated by yeasts from Brazilian gold mines [98].

## 9. Biocontrol

Biocontrol is the use of one or more antagonistically selected microorganisms in agriculture to reduce the presence of pathogen or disease activity. The term “biocontrol” refers to a set of emerging strategies for combating fruit and vegetable diseases, specifically postharvest grape diseases caused by pathogenic fungi such as *Botrytis cinerea*, *Penicillium* spp., and *Aspergillus* spp., the spread of which could result in massive economic losses if not controlled [99]. To reduce the risks and impacts of pesticide use on people’s health and the environment, the EU Directive 2009/128/CE encourages the use of integrated defense and various approaches or techniques, such as non-chemical alternatives to pesticides—including biocontrol strategies—which are also used in food production and storage [99]. Many yeast strains, including epiphytic *Saccharomyces* and endophytic *Metschnikowia*, *Pichia*, and *Hanseniaspora* spp. yeasts, are pathogenic fungi antagonists. If their efficacy is proven, these agents could be used in biocontrol strategies [100]. However, *Saccharomyces* efficacy is frequently inferior to that of non-*Saccharomyces* yeasts and synthetic fungicides. More research is needed to verify this and assess the potential risks to human health. Furthermore, when attempting to combat phytopathogenic fungi, it is important to remember that even if *B. cinerea* can cause “noble putrefaction” on raisins, this increases the concentration of sugars and allows for the production of sweet wines under certain conditions. Understanding the mechanisms and conditions of biocontrol agents’ actions is critical for improving the efficacy of future biocontrol agents because the interaction between pathogen, antagonist, and environment is very complex. The numerous parameters that must be considered further complicate the development of a commercial formulation [99].

## 10. Aquarium Interest

One unusual but useful application of yeast is in the practice of some hobbies. Yeast is frequently used by aquarium hobbyists to produce carbon dioxide (CO_2_) to feed plants in planted aquariums [101]. CO_2_ levels in yeast are more difficult to control than CO_2_ levels in pressurized CO_2_ systems. However, because yeast is inexpensive, it is a popular substitute [101].

## 11. In the Near Future

A new wave of cow-free dairy is hitting the market. In the United States, Perfect Day Inc., a food technology startup company based in Berkeley, California, is using genetically modified fungi to produce milk protein for ice cream at a commercial scale instead of extraction from bovine milk. Pre-commercial companies, such as TurtleTree and Better Milk, are engineering mammary cells to produce human and cow milk in laboratories, although these remain in the early stages of development. These emerging technologies are part of the fourth agricultural revolution that aims to improve food security, sustainability and agricultural working conditions.

Animal agriculture is facing increased scrutiny, especially regarding its environmental impact and animal welfare issues. It is a significant source of greenhouse gas emissions, upwards of 16.5 percent of global emissions. Animal agriculture is also vulnerable to extreme environmental conditions and climate change. Recent flooding in British Columbia, Canada killed well over half a million farm animals and threatened to contaminate the sensitive freshwater ecosystems of the Fraser Valley with stored manure and agricultural chemicals. It is a known risk factor for zoonotic diseases and pandemics, such as H1N1 or the swine flu.

To reduce these risks, “cellular agriculture” was introduced. Cellular agriculture uses cell cultures to produce animal products without raising livestock, hunting or fishing [102,103]. This technology makes biologically equivalent or near-equivalent foods to those produced with animals. One approach is to use advanced fermentation, where yeasts, fungi, and bacteria are genetically modified to produce proteins. This approach is similar to brewing beer, but with highly specialized microorganisms that follow instructions that have been added to their genetic code.

Thirty years ago, the U.S. Food and Drug Administration approved the use of a bioengineered form of rennet enzymes, which are widely used in cheese making and replace the original enzymes, which were harvested from calf stomachs. Today, vats of microorganisms, genetically modified to carry the appropriate calf gene, supply rennet for about 70 percent of cheese made in the U.S. It is functionally identical to the original cheese-making enzymes but it is easier, less costly to produce, and does not rely on mammals.

Professor Tamir Tuller from the Faculty of Engineering at Tel Aviv University and his colleagues is a research group that in the past has developed significant studies and projects to produce vaccines, antibodies, biosensors, and green energy using various organisms such as yeast, bacteria, micro-algae, and even viruses. Professor Tuller and his colleagues set out to conquer a new goal: cow milk. Together with Dr. Eyal Iffergan, Tuller established the startup company Imagindairy, which attempts to produce cow milk from yeast [104]. This startup plans to produce dairy products that will be identical to products that come from animals by introducing the genes that code for milk development in cows to the yeast genome. Could this technological development by Tel Aviv University researchers soon revolutionize the dairy products we consume? The developers believe that in the not-too-distant future, we will be able to buy dairy products in supermarkets that taste and look exactly like the dairy products we currently consume, with one small difference: the dairy products will be made by yeast rather than a cow. In recent years, increased awareness of the damage caused by the dairy industry to the environment and human health and the ethical dilemmas of animal husbandry, biotechnology companies worldwide have been searching for milk substitutes. The goal of Imagindairy is to produce milk with all the important nutritional values of animal milk and the same taste, aroma, and texture that we are all familiar with but without the suffering that cows endure and damage to the environment. These milk and cheese products may actually be much healthier than milk that comes from animals, since they will not contain cholesterol, lactose, or somatic cells.

However, things are not always so simple and clear. Every living creature’s genome contains genes that encode the recipe for constructing the amino acid chains that make up proteins. It does, however, contain information that encodes the complex process known as “gene expression”—the timing and rate at which proteins are generated. Gene expression is the process of converting information stored in “inanimate” DNA into proteins that are the “essence of life” and are found in all living things, from humans to coronaviruses to cow milk. For many years, biotechnology companies have used the gene expression process in order to produce desirable proteins at a low cost. They accomplish this by inserting a gene from one living organism into the genome of another, which will serve as a “factory” for producing the protein encoded in that gene. For many years, this technology has been used to produce medications, vaccines, and energy, and is also used in the food industry. “*Theoretically, we can reach a situation in which we can’t tell the difference between cow’s milk that comes from a cow and cow’s milk that comes from yeast. Even though we know what the genes that encode the proteins for cow’s milk are, those genes are written in the ‘language’ of cow cells, and need to be rewritten in the ‘language’ of yeast. This will make the production of the milk proteins possible in an appropriate, affordable, and efficient way in the yeast cell ‘factory*’.”, recently claimed Professor Tuller, affirming his satisfaction with the result achieved. In fact, in recent weeks reports have circulated that the food tech startup Remilk, a developer of cultured milk and dairy, plans to open the “world’s largest” facility for the production of cow-free milk in Denmark [105,106]. Remilk, a startup which was founded in 2019, uses a yeast-based fermentation process to produce milk proteins that are “chemically identical” to those found in cow-produced milk and dairy products. Remilk recreates milk proteins by inserting the genes that encode them into a single-cell microbe, which they genetically manipulate to express the protein “in an efficient and scalable manner. The product is then powdered. “*We’re making dairy products that are identical to cow-milk products, with the same taste, texture, stretchiness, meltiness, with no cholesterol and no lactose*”, claimed the developers. This news regarding cow-free milk may shock some people, but science cannot be stopped and we cannot censor research. Humans have always imagined things that are unreachable, looking beyond their own horizons. For example, Jules Verne described in his novels adventures that were unthinkable in his time (*Journey to the Center of the Earth* (1864), *Twenty Thousand Leagues Under the Seas* (1870), *Around the World in Eighty Days* (1872)) but which later became possible in part. Scientists are said to be pragmatic, but while this is true (scientific evidence, experimental tests, standardization, reproducibility, experimental demonstrations, hypotheses that must always be verified, etc.), the childish spirit within us is best expressed in the field of science, as curiosity and the desire to discover open scientists to all possibilities. From this vantage point, scientists are unstoppable. Every day, somewhere in the world, revolutionary new scientific discoveries are made in what is colloquially known as “blue-sky” research [107]. Blue-sky thinking and imagination are creative ideas that are not limited by current thinking or beliefs. This way of thinking and curiosity drives scientists to make discoveries and practical applications that will allow humanity to obtain a better quality of life in a sustainable environment. It is still too early to predict whether this research and its subsequent applications will be positive or negative and how our lives will change as a result. The important thing is that the discoveries are used and applied correctly in order to improve our lives and protect the planet on which we live, as Pretorius and Boeke wrote in 2018 “*Explorations have a rich history of resulting in unexpected discoveries and unnticipated applications for the benefit of people and planet*” [107].

## 12. Conclusions

Yeasts have been used by humans since ancient times to make bread, beer, and other products, but since *S. cerevisiae* was sequenced, researchers have been working hard to modify it and make it useful as a system biological model for biotechnological processes. The availability of a complete genome sequence, well-established genetics, and inherent natural adjuvant makes yeasts an ideal model system for biotechnological applications to provide many benefits to humans. This review attempts to provide a rapid overview of the many current and future applications of yeasts, from the production of biofuel to the use of yeasts in bioremediation to remove toxic contaminants and biocontrol to combat phytopathogens; from the production of protein such as insulin to the development of safe vaccines and probiotics; and from the production of food additives to the recent cow-free milk. The main advantages offered by yeast model systems are the well-developed and easily accessible genetic tools, rapid growth, the simple and inexpensive culture media, their manageability, safety, ease of cultivation and reproduction, the fact that many of the cellular and metabolic processes found in higher eukaryotes are conserved in these organisms. Yeasts could be a valid alternative to evaluate a growing number of natural and sustainable applications.

## Data Availability

Not applicable.

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
