# Peer review of "Yeast Genomics and Its Applications in Biotechnological Processes: What Is Our Present and Near Future?"

_jof, 2022, doi:10.3390/jof8070752_

Round 1

Reviewer 1 Report

A review of Yeasts' genomics and their applications in biotechnological pro-2 cesses: what is our present and near future?: A few points deserve attention for further publication. I suggest that it is accepted for publication after the following revisions:

 - The authors could clarify in the abstract of the manuscript the mechanism, advantages, problems, and solutions for the Yeasts' genomics and their applications in biotechnological pro-2 cesses systems.

 - In addition, authors should highlight the advantages / disadvantages of these Yeasts' genomics and their applications in biotechnological pro-2 cesses methods for industrial application and how this information will be addressed in the manuscript.

 - Advantages for the Yeasts' genomics and their applications in biotechnological pro-2 cesses systems: Which methods have advantages? Are they simple methods contribution? When compared with other sustainable techniques? Authors need to leave these clear information to throughout the text and the methods discussed in this manuscript. In addition, this information is needed for the Yeasts' genomics and their applications in biotechnological pro-2 cesses systems contribution protocols are applied on an industrial scale.

 - Problems with the Yeasts' genomics and their applications in biotechnological pro-2 cesses systems: Does this protocol have a significant problem? This discussion could be improved.

 - Additionally, advances in the Yeasts' genomics and their applications in biotechnological pro-2 cesses systems with engineered tailor-made have been performed with other strategies. May open new opportunities. This discussion could be improved.

 - This review had broached interests in the progress and recent applications of the Yeasts' genomics and their applications in biotechnological pro-2 cesses: The main contributions to the accomplishment of this work must be included in the conclusion.

 - Please, check all references according to the author's instructions.

- The manuscript must be formatted according to the journal's standards.

Author Response

Dear Referee,

JoF-1804221.

by Vivian Tullio

 ANSWER TO REFEREE 1 COMMENTS

  • Thank you for your helpful suggestions. I attempted to improve the paper (see the attached paper). However, I don't believe I well comprehended what referee # 1 desired. The comment is a minor revision, but if I understand correctly, I should include methods, advantages, problems, and so on in the abstract and text. However, this would significantly lengthen not only the abstract (which should be no more than 200 words), but also the manuscript. Consequently, it would no longer be a minor revision but rather a major revision. As a result, I believe I erred in my understanding of what I needed to do and how to modify the paper. The goal of this review is to provide an overview of the most important positive applications of yeasts, including those engineered to produce proteins such as insulin, safe vaccines, probiotics, and biofuel, as well as the use of yeasts to produce the most recent milk cows free. There are already comments in the various chapters about any problems, usage, and so on. In any case, I modified the Abstract, the Introduction, and the Conclusions to better specify, as you suggested. I hope that's adequate.
  • Please, check all references according to author’s instructions.

Done. See the Reference section of the manuscript.

  • The manuscript must be formatted according to the journal’s standards.

Done. I used the Jof template.

We look forward to hearing from you.

Yours sincerely,

Prof. Vivian Tullio

Reviewer 2 Report

jof-1804221

  This review is comprehensive on the specifications, classification and applications of yeasts. Also, it well written and introduced. There some comments specially last one as follow:

Comments:

-        Line 60: species instead of organisms

-        Line 66: delete (if)

-        Lines 78-84: add reference(s)

-        Line 115, 177, 195, 199 and others: do not start with abbreviation (S.)

-        Line 129: (Today) is not suitable for a reference 2010

-        Line 163: do not use (recent years) with a reference 2011.

-        Lines 282-285: add reference

-        Lines 295-303: add references

-        Lines 339 to 344 and 347 to 349: and 350 to 354; add references

-        Too many paragraphs were cited from the reference of 62. Author should add other references for these information.

-        Authors should add title (Yeast as biocontrol agent), See these references: https://www.frontiersin.org/articles/10.3389/fmicb.2019.01766/full. doi: 10.3390/foods10071650. DOI: 10.1007/s10482-010-9528-z. https://www.eolss.net/sample-chapters/c10/E5-24-10-05.pdf. https://doi.org/10.1186/s41938-021-00493-4. DOI: 10.1007/978-1-4020-8292-4_10

Author Response

Dear Referee,

Thank you for your answer.  JoF-1804221.

by Vivian Tullio

All changes to the manuscript are indicated in this letter point-by-point. Changes are highlighted by Word-revision software (see attached manuscript).

ANSWER TO REFEREE 2 COMMENTS

Thank you very much for your useful suggestions.

1)Line 60: species instead of organisms.

Done.  

2) Line 66: delete if

Done. 

3)Lines 78-84: add references.

Done ref.n. 6 line 92

4) Lines 115, 177,195, 199, and others: do not start with abbreviation (S.)

Done. I have corrected. I started with the full name.

5) Line 129 (Today) is not suitable for a reference 2010.

I agree. I have deleted Today, see line 144.

6) Line 163: do not use recent years with a reference 2011.

Thank you, I agree. I have modified the sentences (see lines 177-182) and added new references.

7) Lines 282-285. Add references.

Done

8) Lines 295-303. Add references.

Done

9) Lines 339-344; 347-349; 350-354. Add references.

Done

10) Too many paragraphs were cited from n.62. Add other references.

I agree. Done. I added some new references.

11) Add title (Yeasts as biocontrol agents).

Thank you for the suggestion. I added a new title, 9. Biocontrol. I used this title because it seemed more in line with other titles I used.

See lines 496-518.

We look forward to hearing from you.

Yours sincerely,

Prof. Vivian Tullio